# Estimation of Anthocyanins in Whole-Fertility Maize Leaves Based on Ground-Based Hyperspectral Measurements

**Shiyu Jiang, Qingrui Chang \*, Xiaoping Wang, Zhikang Zheng, Yu Zhang**  **and Qi Wang**

College of Nature Resources and Environment, Northwest A&F University, Yangling 712100, China; jiangsy@nwafu.edu.cn (S.J.); wxp4911@nwafu.edu.cn (X.W.); zhengzhikang@nwafu.edu.cn (Z.Z.); yzhhang@nwafu.edu.cn (Y.Z.); wqi@nwafu.edu.cn (Q.W.)
\* Correspondence: changqr@nwsuaf.edu.cn; Tel.: +86-1357-183-5969

**Abstract:** The estimation of anthocyanin (Anth) content is very important for observing the physiological state of plants under environmental stress. The objective of this study was to estimate the Anth of maize leaves at different growth stages based on remote sensing methods. In this study, the hyperspectral reflectance and the corresponding Anth of maize leaves were measured at the critical growth stages of nodulation, tasseling, lactation, and finishing of maize. First-order differential spectra (FD) were derived from the original spectra (OS). First, the spectral parameters highly correlated with Anth were selected. A total of two sensitive bands ($R_\lambda$), five classical vegetation indices ($VI_S$), and six optimized vegetation indices ($VI_C$) were selected from the original and first-order spectra. Then, univariate regression models for Anth estimation (Anth-UR models) and multivariate regression models for estimating anthocyanins (Anth-MR models) were constructed based on these parameters at different growth stages of maize. It was shown that the first-order spectral conversion could effectively improve the correlation between $R_\lambda$, $VI_C$, and Anth, and $VI_C$ are usually more sensitive to Anth than $VI_S$. In addition, the overall performance of Anth-MR models was better than that of Anth-UR models. Among them, Anth-MR models with the combination of three types of spectral parameters ($FD(R_\lambda) + OS\_VI_C + FD\_VI_C/VI_S$) as inputs had the best overall performance. Moreover, different growth stages had an impact on the Anth estimation models, with tasseling and lactation stages showing better results. The best-performing Anth-MR models for these two growth stages were as follows. For the tasseling stage, the best model was the $FD(R_\lambda) + OS\_VI_C + VI_S$-based SVM model, with an $R^2$ of 0.868, RMSE of 0.007, and RPD of 2.19. For the lactation stage, the best-performing model was the $FD(R_\lambda) + OS\_VI_C + FD\_VI_C$-based RF model, with an $R^2$ of 0.797, RMSE of 0.007, and RPD of 2.24. These results will provide a scientific basis for better monitoring of Anth using remote sensing hyperspectral techniques.

**Keywords:** hyperspectral; maize leaves; Anth; classic vegetation index; optimized vegetation index; first-order differential spectra; machine learning algorithm



## 1. Introduction

Anthocyanins are water-soluble flavonoid pigments that are synthesized in the cytoplasm and stored in the vacuoles of the epidermis or mesophyll cells [1–3]. Anthocyanins can provide plants with an adaptive advantage. Numerous studies demonstrate the crucial role anthocyanins play in plant resistance to environmental stresses, including UV radiation, bright light, low temperature, drought, nitrogen and phosphorus deficiency, and pathogens [4–7]. Undoubtedly, the study of absolute or relative amounts of Anth can be used to assess the adaptability of plants to environmental stresses and their growth status. Traditional methods for quantifying Anth have mainly used wet chemical methods, including extraction of Anth from plant tissues using organic solvents such as methanol and ethanol [8,9] and spectrophotometric methods [10]. These methods were expensive, time-consuming, and labor-intensive, and they also destroyed plant tissues, making real-time monitoring of Anth impossible.

In recent years, hyperspectral techniques have been increasingly used to estimate the physiological and biochemical parameters of crops, such as chlorophyll content [11], leaf area index [12], yield [13], water content [13], biomass [14], with promising results. Hyperspectral technology enables rapid, non-destructive, and real-time monitoring of crop growth, which is crucial for managing large agricultural fields [15]. A comparative analysis of the absorption spectra of Anth-free and Anth-containing leaves by Gitelson et al. determined the green band absorption of Anth in leaves near 550 nm [16]. The spectral absorption properties of Anth in specific wavelength bands are manifested in the reflectance spectra of leaves [17]. This allows for a non-destructive estimation of Anth using remote sensing methods. L.J. Janik et al. used visible-near-infrared spectroscopy to predict Anth concentration in red-grape homogenates [18]. Armando Manuel Fernandes et al. used hyperspectral imaging to determine Anth concentration in grape skins [19]. Manuel Larrain et al. used near-infrared spectroscopy to measure three parameters of grape ripeness: sugar (Brix), PH, and Anth concentration [20]. K.R. Manjunath et al. estimated anthocyanin and carotenoids in different species of flowers using hyperspectral data [21], and Liu et al. predicted Anth of prunus cerasifera with hyperspectral data [22].

Vegetation indices (VI) aim to compress a large amount of spectral reflection information into a small number of indicators, enabling the estimation of Anth [23]. Over the past few decades, several $VI_S$ have been developed using spectral curve features, such as the red-green index (R/G) [24], anthocyanin content index (ACI) [25], modified anthocyanin content index (MACI) [26], anthocyanin reflection index (ARI) [16], and modified anthocyanin reflection index (MARI) [27]. These $VI_S$ are closely related to Anth and have been shown to have good accuracy for estimating Anth of the leaves of certain plant species [28,29]. However, leaf structure and pigment composition vary between different plant species. As a result, these $VI_S$ may need to be reparameterized when applied to estimate Anth in different plants.

These classical vegetation indices ($VI_S$) are constructed based on broad bands. However, with the development of hyperspectral technology, narrow-band hyperspectral data can be used to construct optimal vegetation indices ($VI_C$) [13,14]. This involves performing a two-by-two combination of all bands in the hyperspectral data to find the combination with the best correlation to the study target. Several studies have compared $VI_S$ based on broad bands with $VI_C$ based on narrow bands. For example, Tanaka et al. compared $VI_C$ (NDVI, RVI, and DVI)with the traditional broadband vegetation indices (NDVI, EVI, and OSAVI) and found that the model based on the optimal DVI to estimate the LAI of winter wheat had better accuracy [30]. Luo et al. constructed classification models using $R_\lambda$, $VI_S$, and $VI_C$ for distinguishing diseased leaves from healthy leaves of maize and found that the accuracy of the $VI_C$-based classification models was significantly higher than the other two types of models [31]. However, it remains uncertain whether $VI_C$ has better sensitivity to Anth than $VI_S$ and whether the accuracy of the $VI_C$-based Anth estimation models is higher than that of $VI_S$-based models.

Furthermore, it should be noted that most current studies use VI to construct Anth estimation models. However, the impact of spectral noise was ignored in the VI construction process, resulting in the low accuracy of the models. However, it has been shown that first-order spectra can effectively reduce the effect of spectral noise on the target signal [32–35]. In this study, the optimal vegetation index will be constructed using first-order spectra to investigate whether the resulting $FD\_VI_C$ can improve sensitivity to anthocyanins compared to $OS\_VI_C$.

In this paper, we estimated the Anth of maize leaves at individual and whole growth stages using the hyperspectral reflectance of maize leaves at 380–1000 nm, combined with machine learning (ML) methods. The objectives of this study were to: (1) analyze the effects of first-order spectral conversion and different VI on estimating Anth in maize leaves; (2) explore the potential of different Anth estimation models for four critical growth stages in maize. The research results will provide new ideas for rapid, nondestructive, and real-time monitoring of Anth using remote sensing methods.

## 2. Materials and Methods

### 2.1. Study Area Overview

This field trial was conducted in Qinan Village, Xianyang City, Shaanxi Province, China (34°38′N,108°07′E). The study area has a warm temperate, semi-arid, continental monsoon climate, with an average annual temperature of 10.8 °C and an average annual precipitation of 560–600 mm. The region's average elevation is 1000 m, and the experiment was carried out on 22 April 2017, using "Shaanxi Shan 226" maize as the test crop. The study area was divided into 36 small plots and 4 on-farm plots, with a small plot area of 90 m² (9 m × 10 m) and an on-farm plot area of 153 m² (9 m × 17 m).

The soil type in the region is loam, and 3 fertilizers, N, P, and K, were applied. The experiment used 3 treatments, each with 6 levels, and was replicated twice. Only 1 nutrient rate in each treatment was altered. The first treatment involved 6 N levels (0, 30, 60, 90, 120, and 150 kg/ha), and the second treatment had 6 P levels (0, 18.75, 37.5, 56.25, 75, and 93.75 kg/ha), and the final treatment included 6 K levels (0, 20, 40, 60, 80, and 100 kg/ha). For the P and K treatments, the base fertilizer applied was 60 kg/ha N. In the 4 on-farm plots, 4 levels of N at 0, 60, 120, and 180 kg/ha were used. All fertilizers were applied at planting time, and local management practices were followed. Figure 1 shows the location of the study area. Table 1 shows the fertilizer treatment.

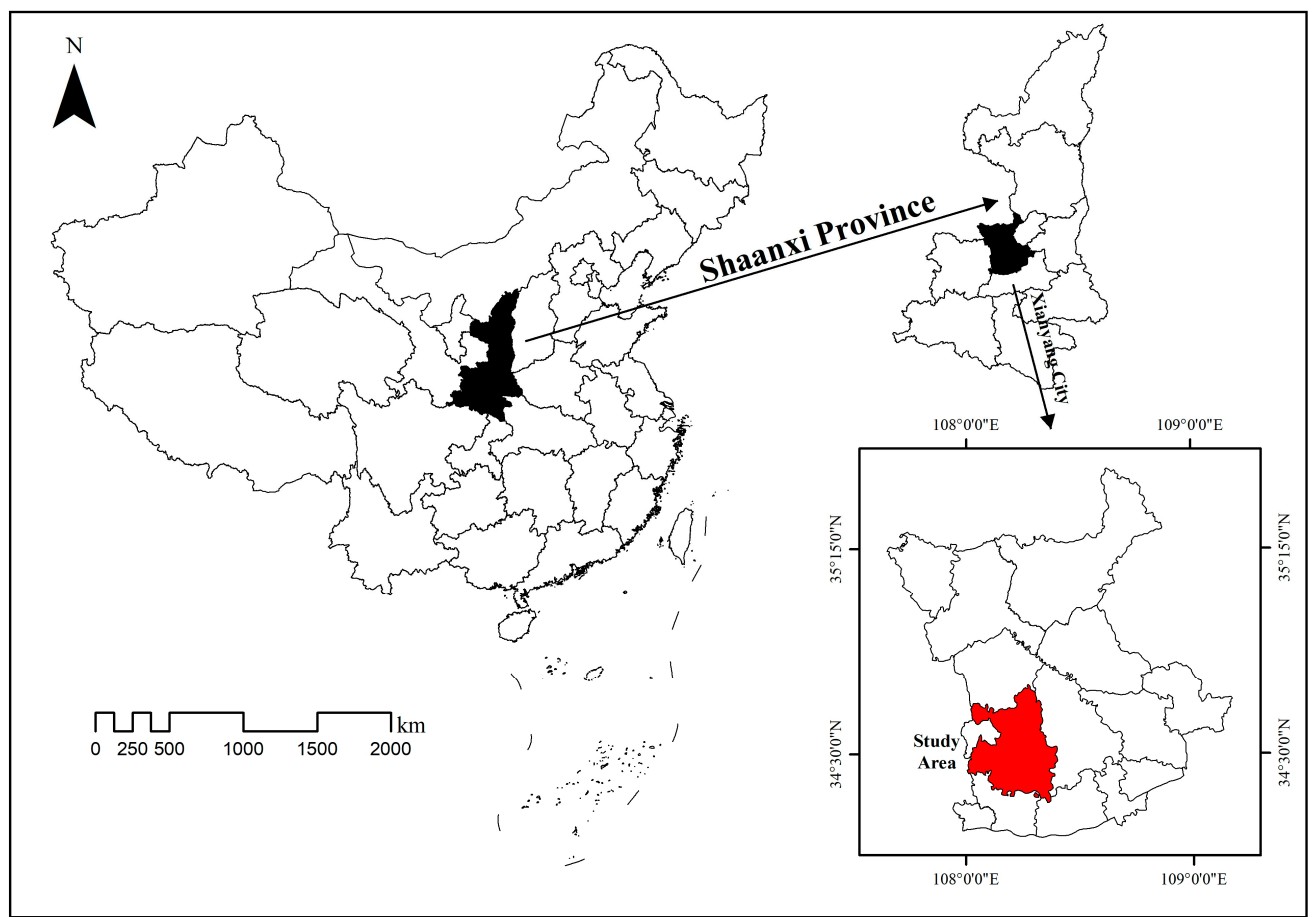

**Figure 1.** Location of the study area.

**Table 1.** Fertilizer Treatment.

| N Treatment Plots | | | | | | P Treatment Plots | | | | | | K Treatment Plots | | | | | | On-Farm Plots | |
|---|---|---|---|---|---|---|---|---|---|---|---|---|---|---|---|---|---|---|---|
| N0 | N1 | N2 | N3 | N4 | N5 | P0 | P1 | P2 | P3 | P4 | P5 | K0 | K1 | K2 | K3 | K4 | K5 | N0 | N6 |
| N0 | N1 | N2 | N3 | N4 | N5 | P0 | P1 | P2 | P3 | P4 | P5 | K0 | K1 | K2 | K3 | K4 | K5 | N2 | N4 |

## 2.2. Data Acquisition

Samples were collected at the maize's nodulation, tasseling, lactation, and finishing stages. For sampling, 2 sample points were selected diagonally for each plot (80 sample points in total), and 3 canopy leaves were picked near the sample points. Figure 2 shows the phenological status of maize at each growth stage.

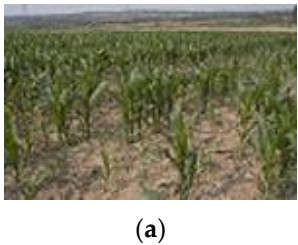 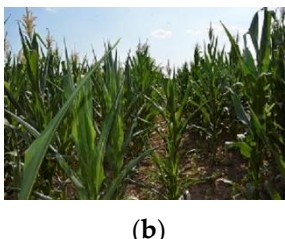 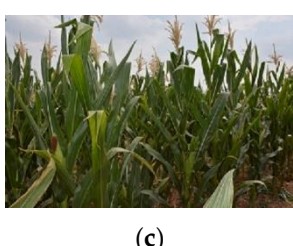 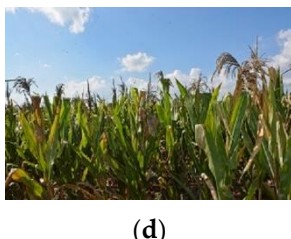

(**a**)            (**b**)            (**c**)            (**d**)

**Figure 2.** Phenological status of maize at each growth stage. (**a**) nodulation stage; (**b**) tasseling stage; (**c**) lactation stage; (**d**) finishing stage.

### 2.2.1. Hyperspectral Data Acquisition

An SVC HR-1024i (Spectra Vista Corporation, Poughkeepsie, NY, USA) portable ground-based spectrometer was used to measure the spectral reflectance of maize leaves indoors. The instrument uses a built-in tungsten lamp as the light source with spectral resolutions of 3.5, 9.5, and 6.5 nm, corresponding to the spectral ranges of 350–1000, 1000–1850, and 1850–2500 nm, respectively. First, the spectral correction was performed using a standard whiteboard. Then, 3 parts of each leaf—the tip, middle, and base—were measured, and 2 spectral curves were obtained for each part. Three leaves were sampled near each sample point, resulting in a total of 18 spectral curves. The average value of the spectral curves was calculated as the sample point's OS. To ensure the data's scientific reliability, a whiteboard calibration was performed every 0.5 h during the experiment. Finally, the spectral resolution was resampled to 1 nm, and Savitzky-Golay smoothing was applied to obtain a smooth reflection spectrum. The spectral resampling operation was performed in the SVC HR-1024i software. The spectral smoothing operation was performed in Unscrambler X 10.4 with a smoothing of 5. Considering the wavelength range of phytochromes, only the spectral range of 380–1000 nm was selected for the study [31]. Figure 3 shows the SVC device.

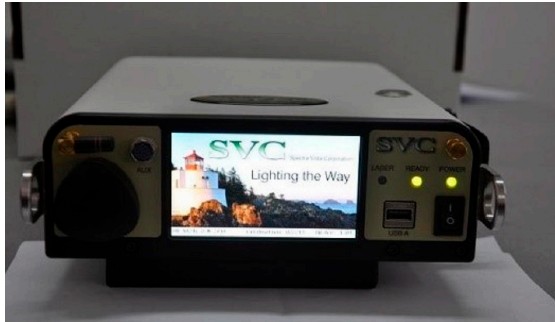

**Figure 3.** The SVC device.

### 2.2.2. Determination of Relative Anthocyanin Content

Dualex Scientific+ (FORCE-A, Orsay, France) was used to non-destructively measure the relative anthocyanin content of maize leaves. The instrument can accurately measure the Anth index and flavonoid index in real time, with a measuring area of 5 mm$^2$. A detailed description of the instrument can be found in the publication by Goulas [36]. In this experiment, the relative Anth content of different parts of the leaf (tip, middle, and base) was measured, with 2 Anth indices obtained for each part. Three leaves were measured per sample site, resulting in a total of 18 Anth indices per site. The average value

of these indices was then calculated to determine the actual relative Anth content of each sample site.

*2.3. Methods*

2.3.1. Spectral Transformation (ST)

In this study, we performed spectral transformations of FD to extract sensitive bands and VI. Compared to OS, FD can effectively reduce the effect of noise on the target signal, highlight spectral feature information, and amplify the details of the OS curves [32–35].

2.3.2. Vegetation Indices (VI)

The vegetation index refers to the combined operation of normalization, ratio, and differences in the reflectance of related bands [37]. By combining bands, the index can reduce the influence of sensors and background on the target, enhance the linear response to the target, and make better use of spectral information while reducing data dimensionality [26]. Liu et al. previously summarized $VI_S$ for estimating anthocyanin content and applied them to maize [38]. In this study, we constructed 5 $VI_S$ (R/G [24], ACI [25], MACI [26], ARI [16], and MARI [27]) based on Liu's [38]. Hyperspectral data contains a vast amount of information due to its numerous bands. To account for this, we operated on all possible 2-2 combinations of bands to find the optimal combination for constructing 3 $VI_C$ (DVI, RVI, and NDVI) to estimate Anth. The calculation of various vegetation indices is presented in Table 2, and the optimal band combinations for the 3 $VI_C$ are detailed in the Results section.

**Table 2.** Classic Vegetation Indices.

| VIs | Equations | References |
|:---:|:---:|:---:|
| R/G | $R_{\lambda Red}/R_{\lambda green}$ | [11] |
| ACI | $R_{\lambda green}/R_{\lambda NIR}$ | [12] |
| MACI | $R_{\lambda NIR}/R_{\lambda green}$ | [13] |
| ARI | $R^{-1}_{\lambda green}/R^{-1}_{\lambda rededge}$ | [8] |
| MARI | $\left(R^{-1}_{\lambda green}/R^{-1}_{\lambda rededge}\right)R_{NIR}$ | [14] |

Note: Waveband range: λRed: 660–680 nm; λgreen: 540–560 nm; λrededge: 700–760 nm; λNIR: 760–800 nm; R indicates the average reflectance of the band range.

2.3.3. Random Forest (RF) Regression

Bagging predictors is an ML technique that aggregates the results of multiple predictors computed in parallel to obtain a combined prediction [39]. RF is a specific algorithmic implementation of the Bagging method. It is a tree-structured classifier that consists of multiple decision trees using randomly selected features to classify samples into different leaf nodes for classification or regression. The function TreeBagger in Matlab can implement the random forest algorithm. The TreeBagger function integrates a set of decision trees that can be used for classification or regression prediction. The "ntree" parameter represents the number of decision trees, and the "mtree" parameter represents the minimum number of leaves. "ntree" and "mtree" are the 2 key parameters that control the performance and complexity of the random forest model. In this study, "ntree" was set to 500 and "mtry" was set to 1/3 of the sample size [40]. In regression prediction, the final output of the RF is the average of the results of multiple decision trees [40]. RF models were implemented in MATLAB R2019b [41].

2.3.4. Support Vector Regression (SVR)

SVR is an ML algorithm developed based on statistical learning theory [42]. Its core principle is to use kernel functions to map the data into a high-dimensional feature space and construct the optimal hyperplane in that space [43]. This approach has shown excellent generalization ability and high prediction accuracy in previous studies [44]. When using the RBF kernel, 2 key parameters must be taken into account: the penalty

parameter (C) and the kernel parameter ($\gamma$). The parameter "C" is a trade-off between the smoothness of the decision surface and the classification deviation. The "$\gamma$" parameter defines the degree of influence of a single training sample, which means that the smaller the parameter, the greater the influence. In this paper, the optimal penalty parameter (C) and the kernel parameter ($\gamma$) are obtained using the grid search method. The parameters C and $\gamma$ are optimized in the range of $[10^{-2}, 10^{-1}, 1, 10, 100]$ and $[10^{-4}, 10^{-3}, 10^{-2}, 10^{-1}, 1, 10]$, respectively. The procedure was implemented in MATLAB R2019b [41].

### 2.3.5. Backward Propagation Neural Network (BPNN)

BPNN is a commonly used method for the estimation of vegetation physiological and biochemical parameters [45–48]. The network mainly consists of an input layer, a hidden layer, and an output layer, with the hidden layer passing important information between the input and output layers [49]. BPNN mainly involves 2 processes: forward transmission of information and backward transmission of errors, where the core of the method is to adjust the weights of neurons by backpropagating the errors [50,51]. In this study, the input layer neuron parameter "m" and the output layer neuron parameter "n" of BPNN are determined by the number of independent and dependent variables. The number of nodes in the hidden layer (hiddennum) can be determined using the empirical Formula (1), where the parameter "a" is a constant in the range of 1 to 10 integers. In this BPNN model, "tansig" is chosen as the activation function, "pureline" is selected as the transfer function, and "trainlm" is chosen as the training function. The BPNN models were implemented using MATLAB R2019b [41].

$$\text{hiddennum} = \sqrt{m + n} + a \tag{1}$$

### 2.3.6. Evaluation Metrics for Model Accuracy

The measured data for each growth stage were sorted in ascending order by anthocyanin content, stratified, and randomly sampled in a 3:1 ratio before modeling. For each growth stage, a calibration set of 60 samples and a validation set of 20 samples were obtained. The data from all 4 growth stages were pooled together as a single dataset for the entire growth stage ($S_{all}$). Models were trained on the calibrated dataset and evaluated on an independent validation dataset.

The performance of the Anth estimation model was evaluated using 3 metrics: determination coefficient ($R^2$), root-mean-squared error (RMSE), and relative prediction deviation (RPD). In this paper, the model with the highest $R^2$ and the lowest RMSE was considered the best 1. On this basis, the values of RPD are referenced: RPD < 1.0 indicates very poor models and are not considered; 1.0 < RPD < 1.4 indicates poor models, with only high and low values distinguishable; 1.4 < RPD < 1.8 indicates fair models that can be used for evaluation and correlation; 1.8 < RPD < 2.0 indicates good models with possible quantitative prediction; 2.0 < RPD < 2.5 indicates very good quantitative models; RPD > 2.5 indicates excellent model [52]. RPD is calculated using Formula (1).

$$\text{RPD} = \text{SD}/\text{RMSE} \tag{2}$$

## 3. Results

### 3.1. Statistical Analysis of Anth

The statistics of Anth for each growth stage are shown in Table 3. The data shows that Anth of the calibration and validation sets has similar statistical characteristics. Anth varied from 0.045 to 0.152 $\mu g/cm^2$, and the coefficient of variation ranged from 15.390% to 25.506%. The average value of Anth showed a pattern of decreasing, then increasing, and finally decreasing again during the growth stages of maize, through the nodulation, tasseling, lactation, and finishing stages.

**Table 3.** Statistics of Anth measurements at each growth stage of maize($S_{all}$ presents the whole growth stage; the same as below).

| Dataset | Growth Stage | Sample Numbers | Range | Mean | Standard Deviation | Coefficient of Variation/% |
|---|---|---|---|---|---|---|
| Calibration set | nodulation | 60 | 0.064–0.146 | 0.105 | 0.016 | 15.390 |
| | tasseling | 60 | 0.053–0.152 | 0.082 | 0.017 | 20.859 |
| | lactation | 60 | 0.071–0.146 | 0.098 | 0.016 | 15.985 |
| | finishing | 60 | 0.047–0.128 | 0.076 | 0.017 | 21.998 |
| | $S_{all}$ | 240 | 0.045–0.152 | 0.090 | 0.020 | 22.033 |
| Validation set | nodulation | 20 | 0.073–0.144 | 0.105 | 0.017 | 16.230 |
| | tasseling | 20 | 0.053–0.114 | 0.082 | 0.016 | 19.380 |
| | lactation | 20 | 0.078–0.141 | 0.098 | 0.017 | 16.887 |
| | finishing | 20 | 0.045–0.132 | 0.076 | 0.020 | 25.506 |
| | $S_{all}$ | 80 | 0.047–0.146 | 0.090 | 0.020 | 22.033 |

### 3.2. Characteristics of Reflectance Spectra

The spectral characteristics of maize leaves were generally consistent trends across the four growth stages, as shown in Figure 4a. In the visible range, a reflection peak was observed near the 550 nm band, and a "red edge" was observed in the 680–760 nm band. In the near-infrared range, each growth stage formed a high reflectance platform in the 780–1000 nm range, with reflectance values above 0.4. These are common features of the green plant spectrum. These spectral features are primarily influenced by leaf pigments in the visible range and by leaf cell structure in the near-infrared range.

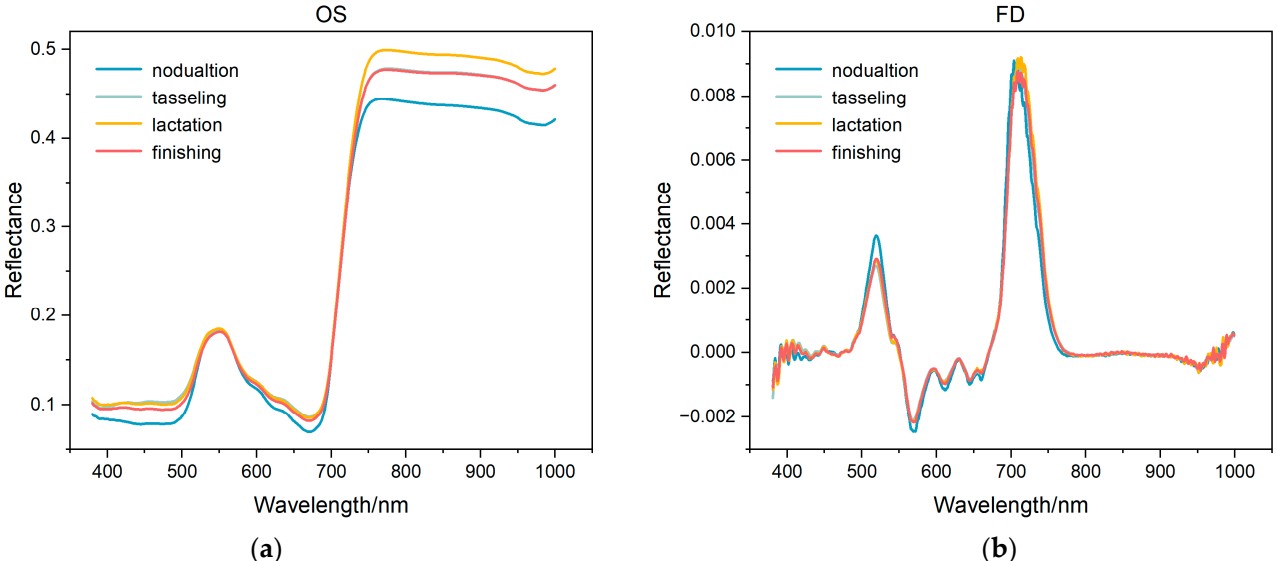

**Figure 4.** (**a**) Original spectra of maize leaves at each growth stage; (**b**) First-order differential reflectance of maize leaves at each growth stage.

The spectral characteristics of FD of maize leaves at the four growth stages are presented in Figure 4b. FD of the nodulation, tasseling, lactation, and finishing stages showed maximum values in the green light band range at band positions 520, 519, 519, and 520 nm, respectively, and in the red light band range at band positions 704, 711, 713, and 709 nm, respectively. This finding suggests that these band positions in the OS are where the reflectance rises and changes the fastest, indicating that FD can amplify the spectral features of OS.

### 3.3. Correlation between Anth and Spectral Reflectance

#### 3.3.1. Correlation between Anth and Spectrum

The correlation between Anth and OS is shown in Figure 5a. At the nodulation and finishing stages, Anth and OS exhibited a positive correlation in the 380–740 nm and 380–745 nm band ranges, respectively, and a negative correlation in other band ranges. At the tasseling and lactation stages, Anth and OS were positively correlated in the 380–1000 nm band. The number of bands that showed a correlation coefficient passing the 0.01 significance level test were, from lowest to highest, the finishing stages, nodulation, lactation, and tasseling stages. The highest correlation coefficients were 0.690, 0.723, 0.722, and 0.427 for the nodulation, tasseling, lactation, and finishing stages, respectively, corresponding to the band positions of 698, 553, 560, and 712 nm. These band positions are concentrated in the visible range where OS forms reflection peaks and "red edges", indicating the sensitivity of these bands to Anth.

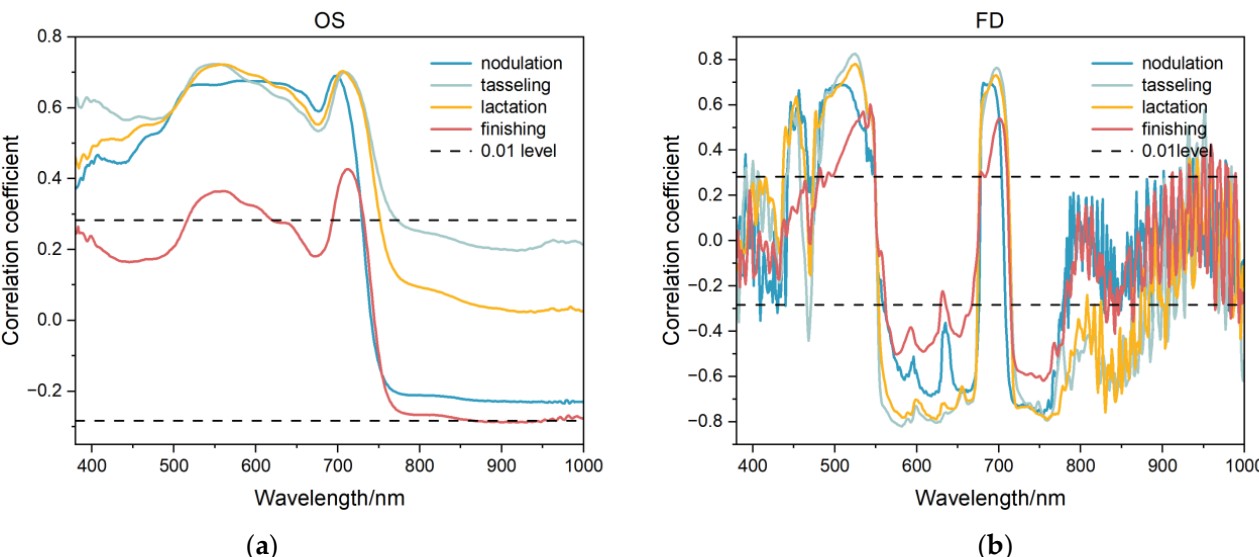

**(a)**  **(b)**

**Figure 5.** (**a**) Correlation between original spectra and anthocyanins of maize leaves at each growth stage; (**b**) Correlation between First-order differential reflectance and anthocyanins of maize leaves at each growth stage.

The correlation of Anth with the FD is shown in Figure 5b. The highest absolute correlation coefficient values were 0.770, 0.825, 0.790, and 0.615 for the nodulation, tasseling, lactation, and finishing stages, respectively, corresponding to the band positions of 754, 524, 759, and 755 nm. In all four growth stages, the maximum correlation coefficients between FD and Anth were higher than those between OS and Anth. This finding suggests that FD can enhance the sensitivity to Anth.

#### 3.3.2. Correlation between Anth and VI$_S$

According to Table 2, 5 VI$_S$ with good correlation to Anth were established. Figure 6 depicts the correlation between Anth and the 5 VI$_S$ constructed from the OS of maize leaves at different growth stages. The correlation coefficients of these five VI$_S$ with Anth passed the significance test at the 0.01 level for all growth stages. At the nodulation, lactation, and S$_{all}$ stages, the ACI had the highest correlation with Anth, with correlation coefficients of 0.74, 0.77, and 0.61, respectively. At the tasseling and finishing stages, R/G had the highest correlation with Anth, with correlation coefficients of −0.77 and −0.46.

In addition, MACI, ARI, and MARI also performed well. At the nodulation, tasseling, and lactation stages, MACI had correlation coefficients with Anth greater than 0.7, while ARI had coefficients greater than 0.6. MARI had coefficients greater than 0.7 at the nodulation and lactation stages and greater than 0.69 at the tasseling stage. At the finishing stage, however, the correlation coefficients between all VI$_S$ and Anth were less than 0.5. At the

$S_{all}$ stage, all four $VI_S$, except for R/G, had correlation coefficients greater than 0.5. Overall, all $VI_S$ improved the response to Anth compared to OS.

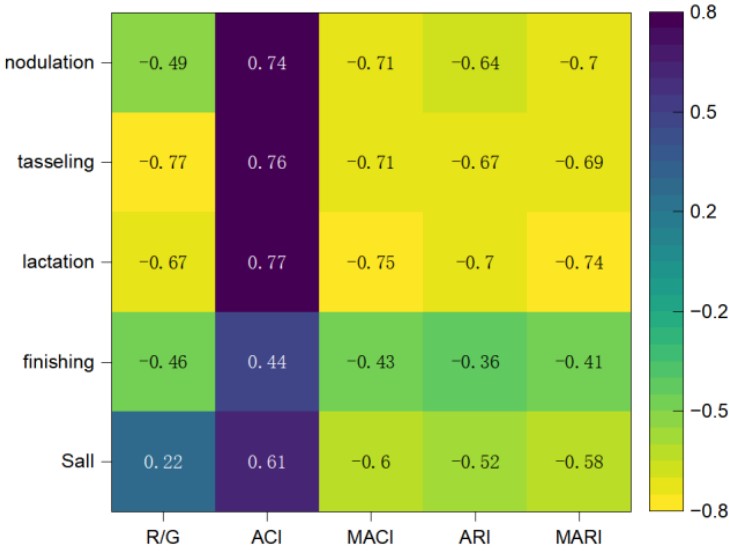

**Figure 6.** The Correlation coefficient between $VI_S$ and Anth at each growth stage.

### 3.3.3. Correlation between Anth and $VI_C$

DVI, RVI, and NDVI were calculated for each growth stage in the 380–1000 nm band range based on any two-band combination of the OS and FD of maize leaves. The contour maps of the correlation coefficients between $VI_C$ and Anth constructed based on the OS are shown in Figure 7. DVI showed the best response to Anth at the nodulation, tasseling, and $S_{all}$ stages, with maximum correlation coefficients of 0.770, 0.826, and 0.718, corresponding to the band positions $(R_{753}, R_{755})$, $(R_{525}, R_{524})$, and $(R_{747}, R_{879})$, respectively. RVI showed the best response to Anth at the lactation and finish stages, with maximum correlation coefficients of 0.793 and 0.694, corresponding to the band positions $(R_{827}, R_{834})$ and $(R_{513}, R_{700})$, respectively.

The contour maps of the correlation coefficients between $VI_C$ and Anth constructed based on the FD are shown in Figure 8, in which the white part is caused by the non-existence of the corresponding FD_RVI and FD_NDVI. RVI showed the best response to Anth at the nodulation and finish stages, with maximum correlation coefficients of 0.815 and 0.687, corresponding to the band positions $(R_{997}, R_{755})$ and $(R_{543}, R_{993})$, respectively. NDVI showed the best response to Anth at the tasseling, lactation, and $S_{all}$ stages, with maximum correlation coefficients of 0.859, 0.854, and 0.730, corresponding to the band positions $(R_{737}, R_{759})$, $(R_{756}, R_{759})$, and $(R_{676}, R_{772})$, respectively.

### 3.4. Univariate Regression Model for Anth Estimation (Anth-UR)

Based on the principle of the maximum correlation coefficient, two sensitive bands of OS and FD, and 11 optimal VI, including R/G, ACI, MACI, ARI, MARI, OS_DVI, OS_RVI, OS_NDVI, FD_DVI, FD_RVI, and FD_NDVI, were selected. Then, 13 Anth-UR were constructed for each growth stage. The $R^2$, RMSE, and RPD of the 13 models for each growth stage are shown in Figure 9.

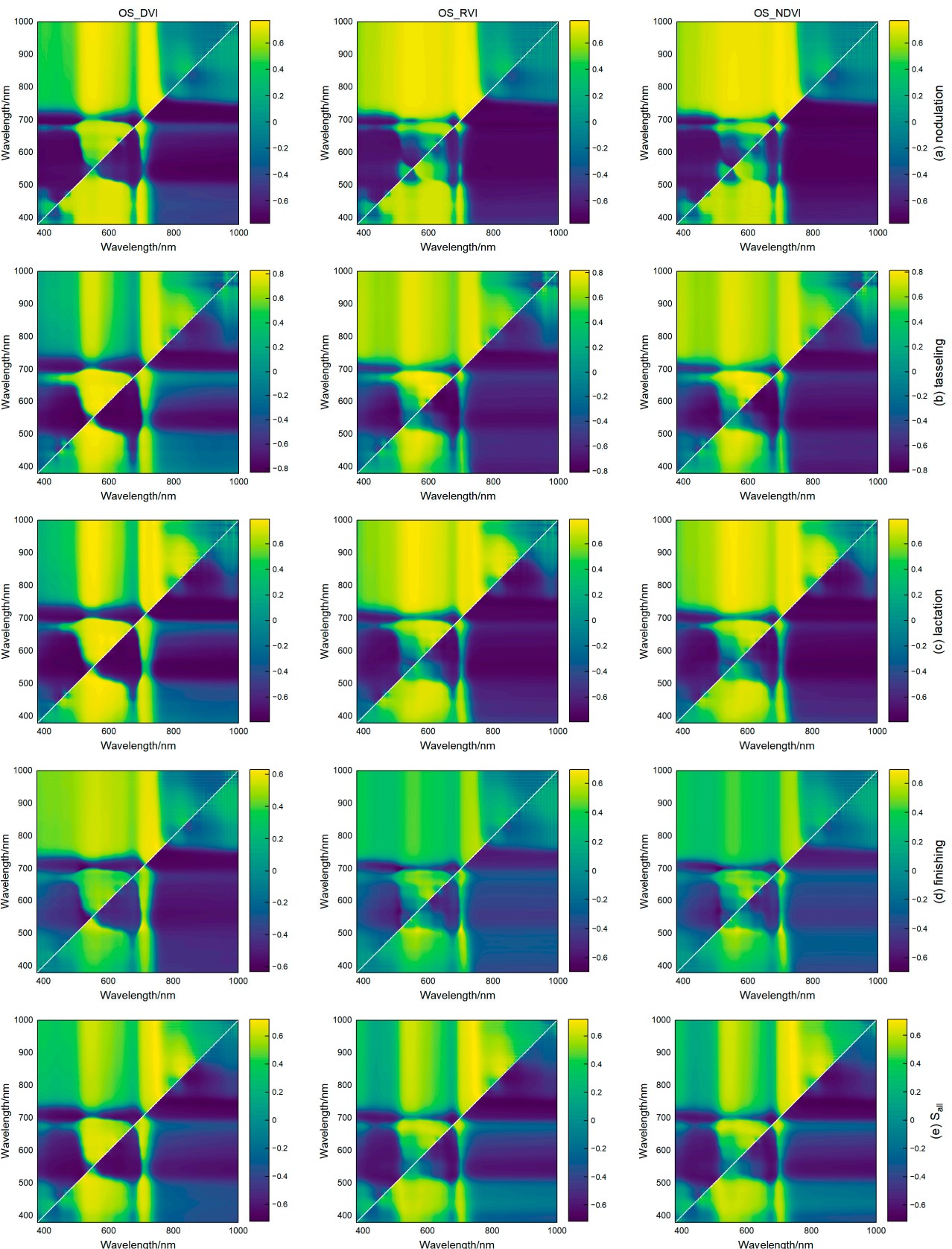

**Figure 7.** Contour maps of the Correlation coefficient between OS_VI$_C$ and Anth at each growth stage. (**a**) the nodulation stage; (**b**) the tasseling stage; (**c**) the lactation stage; (**d**) the finishing stage; (**e**) the S$_{all}$ stage.

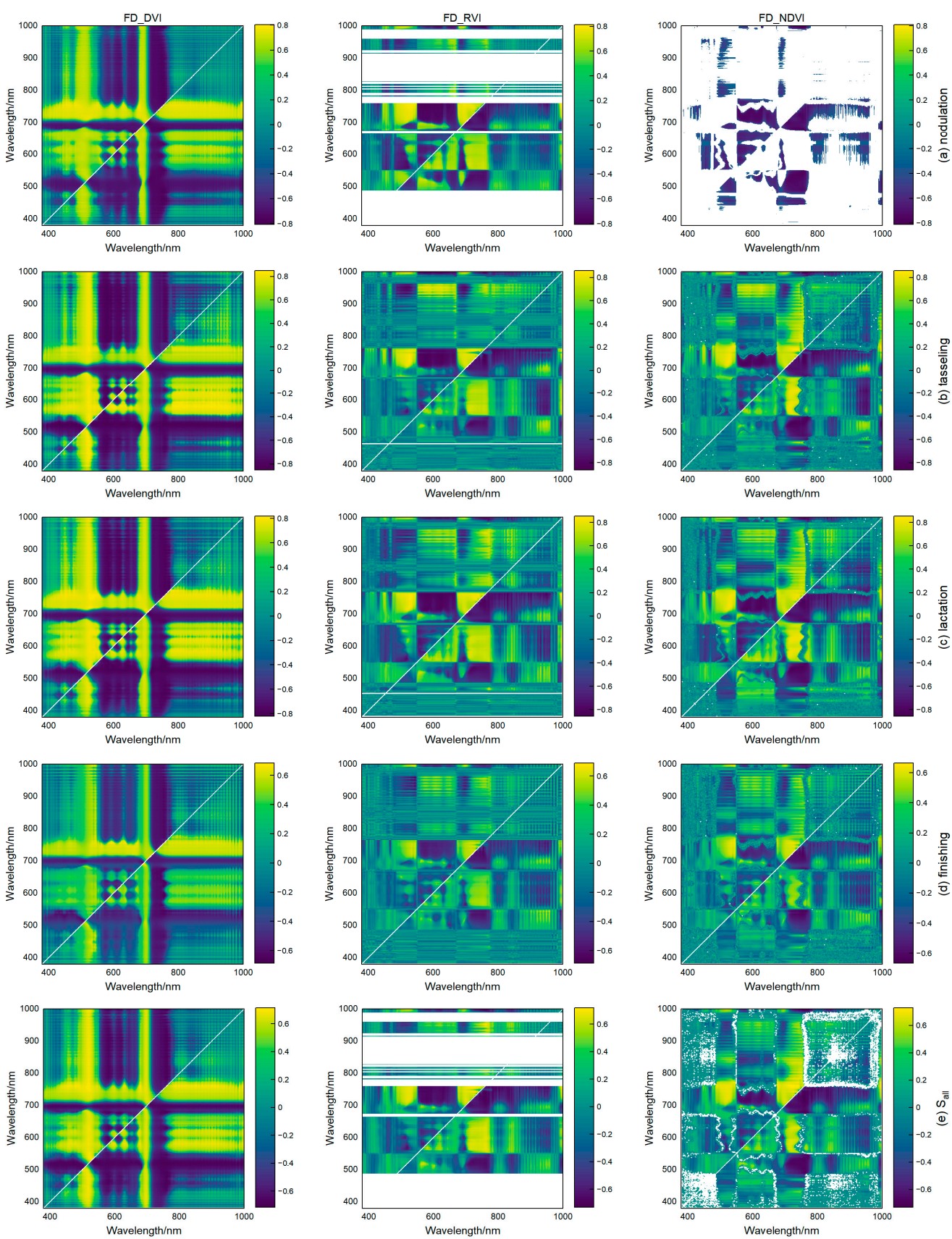

**Figure 8.** Contour maps of the Correlation coefficient between FD_VI$_C$ and Anth at each growth stage. (**a**) the nodulation stage; (**b**) the tasseling stage; (**c**) the lactation stage; (**d**) the finishing stage; (**e**) the S$_{all}$ stage.

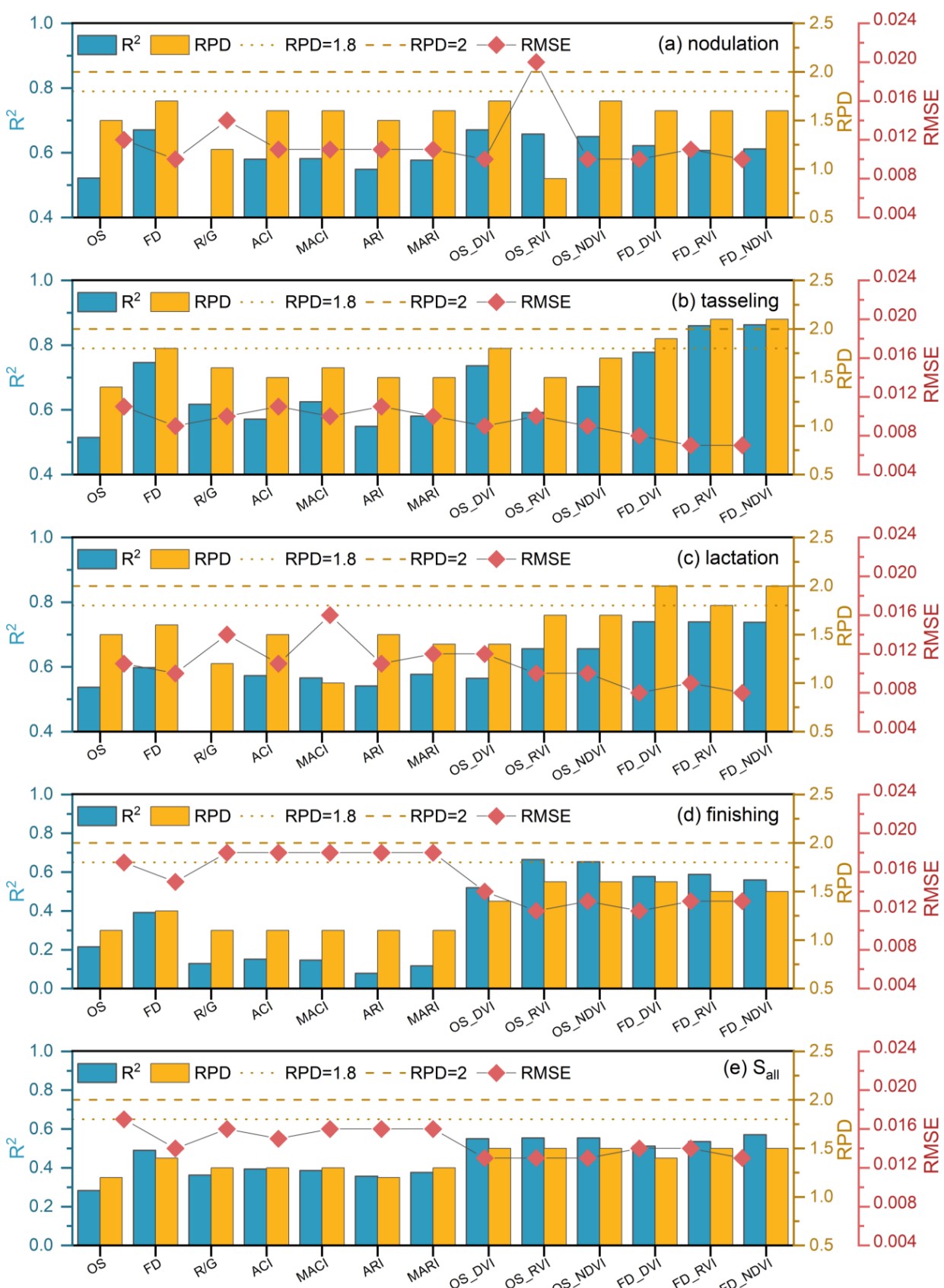

**Figure 9.** Anth-UR Models accuracy parameters for each growth stage. (**a**) represents the nodulation stage; (**b**) represents the tasseling stage; (**c**) represents the lactation stage; (**d**) represents the finishing stage; (**e**) represents the S$_{all}$ stage.

For the nodulation stage, the best-performing models were the FDS ($R_{754}$)-based model and the OS_DVI ($R_{753}$, $R_{755}$)-based model, both with an $R^2$ of 0.671 and an RMSE of 0.010 for the validation set. For the tasseling stage, the FD_NDVI ($R_{737}$, $R_{759}$)-based model had the highest $R^2$ of 0.863 and the lowest RMSE of 0.007. For the lactation stage, the FD_RVI ($R_{756}$, $R_{759}$)-based model had the highest $R^2$ of 0.739, and the FD_NDVI ($R_{756}$, $R_{759}$)-based model had the lowest RMSE of 0.008. For the finish stage, the best-performing model was based on OS_RVI ($R_{513}$, $R_{700}$), with an $R^2$ of 0.665 and an RMSE of 0.012. For the $S_{all}$ stage, the best-performing model was based on FD_NDVI ($R_{676}$, $R_{772}$), with an $R^2$ of 0.571 and an RMSE of 0.013.

The FD($R_\lambda$)-based models performed better in all growth stages than the OS($R_\lambda$)-based models, suggesting that spectral conversion improves the response to Anth. For the models constructed based on $VI_S$, the models based on R/G performed better only in the tasseling stage. The models based on ACI, MACI, ARI, and MARI performed better than the OS($R_\lambda$)-based models for all four growth stages, except for the finishing stage. The models based on ACI and MACI outperformed those based on ARI and MARI overall. The models based on $VI_C$ performed better at all growth stages than the OS($R_\lambda$)-based models. Moreover, the overall accuracy of FD_$VI_C$-based models was higher than that of OS-$VI_C$-based models at the nodulation, tasseling, lactation, and finish stages.

The RPD histograms for the five growth stages are shown in Figure 9. The RPD values were less than 1.8 for all models at the nodulation, finishing, and $S_{all}$ stages, indicating that these models were not recommended. For the tasseling stage, the FD($R_{524}$)-based, OS_DVI($R_{525}$, $R_{524}$)-based, and FD_DVI($R_{525}$, $R_{435}$)-based models with $1.8 < RPD < 2.0$ can be used as quantitative models. Meanwhile, the FD_RVI($R_{745}$, $R_{754}$)-based and FD_NDVI($R_{737}$, $R_{759}$)-based models with $RPD > 2.0$ can be considered as very stable models for quantitative estimation of maize's Anth at the tasseling stage. For the lactation stage, the FD_RVI($R_{756}$, $R_{759}$)-based, FD_DVI($R_{889}$, $R_{772}$)-based, and FD_NDVI($R_{756}$, $R_{759}$)-based models with $1.8 < RPD < 2.0$ can be used as quantitative models.

### 3.5. Multiple Regression Model for Anth Estimation (Anth-MR)

#### 3.5.1. Anth-MR Model Based on FD($R_\lambda$) + $VI_S$\OS_$VI_C$\FD_$VI_C$

Since both the correlation coefficients of FD($R_\lambda$) with Anth and the performance of the Anth-UR models were better than OS($R_\lambda$), FD($R_\lambda$) combined with $VI_S$, OS_$VI_C$, and FD_$VI_C$ were selected as independent variables to construct Anth-MR models. The model parameters for each growth stage are shown in Figure 10.

For the nodulation stage, the best-performing RF model was constructed based on the combination of FD($R_\lambda$) + FD_$VI_C$ spectral parameters. The best-performing SVM and BPNN models were both constructed based on FD($R_\lambda$) + $VI_S$, and the SVM model had the highest accuracy with $R^2$ of 0.682, RMSE of 0.009, and RPD of 1.81.

For the tasseling and lactation stages, the best-performing RF, SVM, and BPNN models were constructed based on FD($R_\lambda$) + FD_$VI_C$. The SVM model had the highest accuracy for the tasseling stage with $R^2$ of 0.853, RMSE of 0.007, and RPD of 2.2. The best-performing model for the lactation stage was the BPNN model with $R^2$ of 0.773, RMSE of 0.008, and RPD of 2.2.

For the finish stage, the best-performing RF, SVM, and BPNN models were all constructed based on FD($R_\lambda$) + OS_$VI_C$. The RF model had the highest precision with $R^2$ of 0.610, RMSE of 0.013, and RPD of 1.52.

For the $S_{all}$ stage, the best-performing RF and BPNN models were constructed based on FD($R_\lambda$) + FD_$VI_C$, and the best-performing SVM model was constructed based on FD($R_\lambda$) + OS_$VI_C$. The BPNN model had the highest accuracy with $R^2$ of 0.598, RMSE of 0.013, and RPD of 1.54.

The RPD histograms for the five growth stages are presented in Figure 10. For the nodulation stage, only the FD($R_\lambda$) + $VI_S$-based SVM, $1.8 < RPD < 2.0$, could be used as a quantitative Anth-MR model of the nodulation stage of maize. For the tasseling stage, six models with $1.8 < RPD < 2.0$ are considered suitable quantitative models, including the

FD($R_\lambda$) + FD_VI$_C$-based RF model, the FD($R_\lambda$) + OS_VI$_C$-based and FD($R_\lambda$) + VI$_S$-based SVM models, and all models of BPNN. While the FD($R_\lambda$) + FD_VI$_C$-based SVM model with RPD > 2.0 is regarded as a highly stable quantitative Anth-MR model of the tasseling stage. For the lactation stage, three models with 1.8 < RPD < 2.0 can be considered as quantitative models, including the FD($R_\lambda$) + OS_VI$_C$-based RF model, the FD($R_\lambda$) + OS_VI$_C$-based SVM, and FD($R_\lambda$) + FD_VI$_C$-based SVM. There are three models with RPD > 2.0, including the FD($R_\lambda$) + FD_VI$_C$-based RF model, the FD($R_\lambda$) + OS_VI$_C$-based BPNN model, and the FD($R_\lambda$) + FD_VI$_C$-based BPNN model, can be considered as very stable quantitative Anth-MR model of the lactation stage. For the finishing and S$_{all}$ stages, all models, RPD < 1.8, are not recommended as Anth-MR.

### 3.5.2. Anth-MR Model Based on FD($R_\lambda$) + OS_VI$_C$ + FD_VI$_C$\VI$_S$

The performance of the model based on the spectra combination of $R_\lambda$ + VI$_S$ + VI$_C$ was better than that of the VI$_S$-based or VI$_C$-based models in Lili's study [31]. Therefore, the spectral parameter combinations of FD($R_\lambda$) + OS_VI$_C$ + VI$_S$ and FD($R_\lambda$) + OS_VI$_C$ + FD_VI$_C$ were used as independent variables to construct the Anth-MR models. These models' accuracy parameters for each growth stage are shown in Figure 11. For the nodulation and finishing stages, the only FD($R_\lambda$) + OS_VI$_C$ + VI$_S$-based SVM model, 1.8 < RPD < 2.0, could be used as a model for quantitative estimation of Anth with $R^2$ of 0.681 and 0.711, RMSE of 0.009 and 0.011, and RPD of 1.81 and 1.82, respectively. For the tasseling stage, all six models had good accuracy, with $R^2$ > 0.75 and RMSE < 0.008. Except for the two RF models, the remaining models, with RPD > 2.0, can be considered very stable quantitative Anth-MR models of the tasseling stage of maize. The best model was the FD($R_\lambda$) + OS_VI$_C$ + VI$_S$-based SVM model, with an $R^2$ of 0.868, RMSE of 0.007, and RPD of 2.19. For the lactation stage, all six models demonstrated good accuracy, with an $R^2$ greater than 0.7 and an RMSE less than 0.009. However, the FD($R_\lambda$) + OS_VI$_C$ + VI$_S$-based SVM model is not recommended for quantitative estimation of Anth. On the other hand, the remaining five models, with an RPD greater than 2.0, can be considered stable models for quantitative estimation of Anth. Among these, the best-performing model was the FD($R_\lambda$) + OS_VI$_C$ + FD_VI$_C$-based RF model, achieving an $R^2$ of 0.797, an RMSE of 0.007, and an RPD of 2.24. For the S$_{all}$ stage, All models with RPD < 1.8 are not recommended for quantitative estimation of Anth.

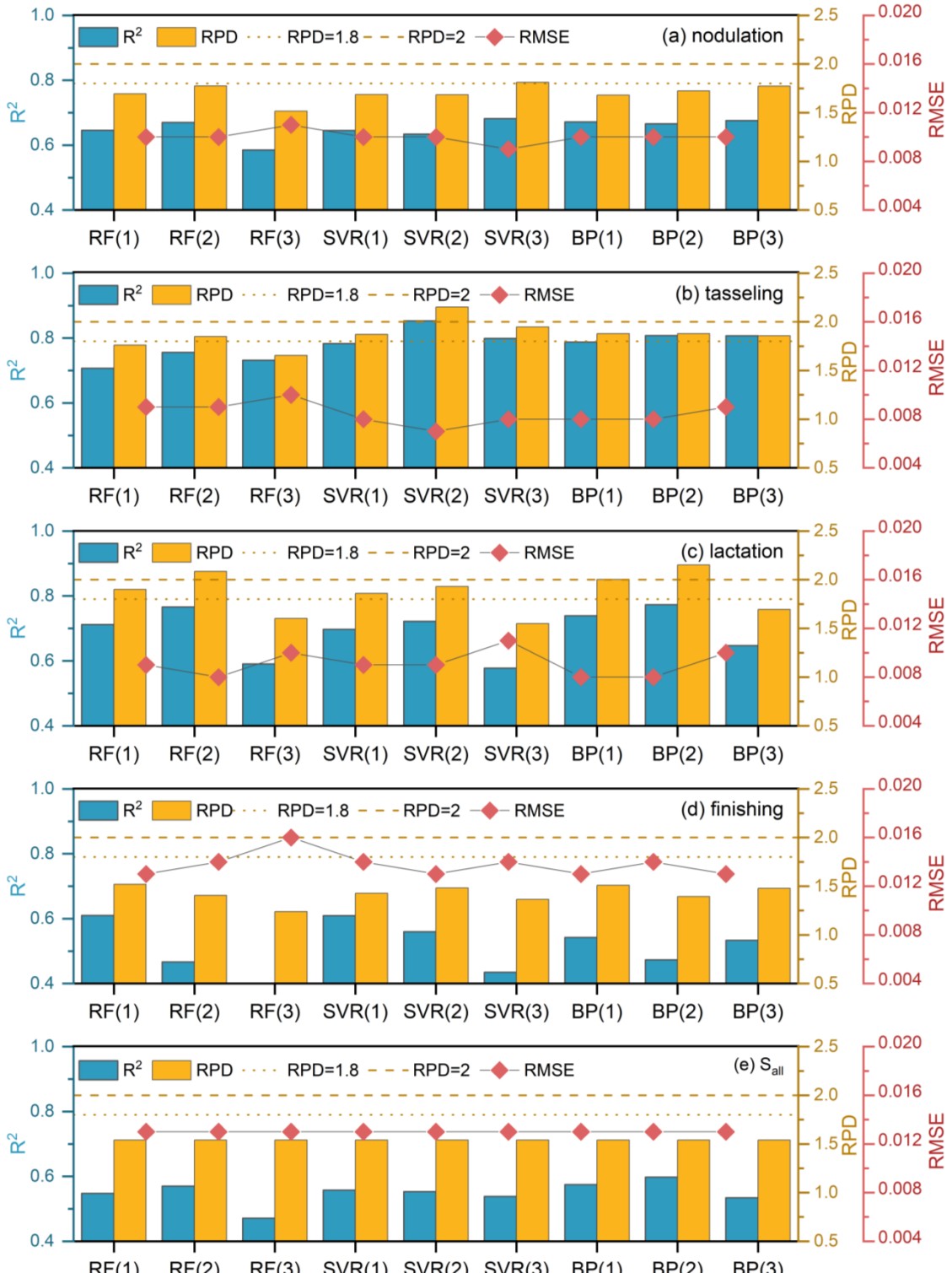

**Figure 10.** Model accuracy parameters for each growth stage. (**a**): the nodulation stage; (**b**): the tasseling stage; (**c**): the lactation stage; (**d**): represents the finishing stage; (**e**): represents the $S_{all}$ stage; (1) represents the model parameters $FD(R_\lambda) + OS\_VI_C$ model using Random Forest: RF(1), Support Vector Regression: SVR(1) and Back Propagation Neural Network: BPNN(1); (2) represents the model parameters $FD(R_\lambda) + FD\_VI_C$ model using Random Forest: RF(2), Support Vector Regression: SVR(2) and Back Propagation Neural Network: BPNN(2); (3) represents the model parameters $FD(R_\lambda) + VI_S$ model using Random Forest: RF(3), Support Vector Regression: SVR(3) and Back Propagation Neural Network: BPNN(3). BP in the figure represents BPNN.

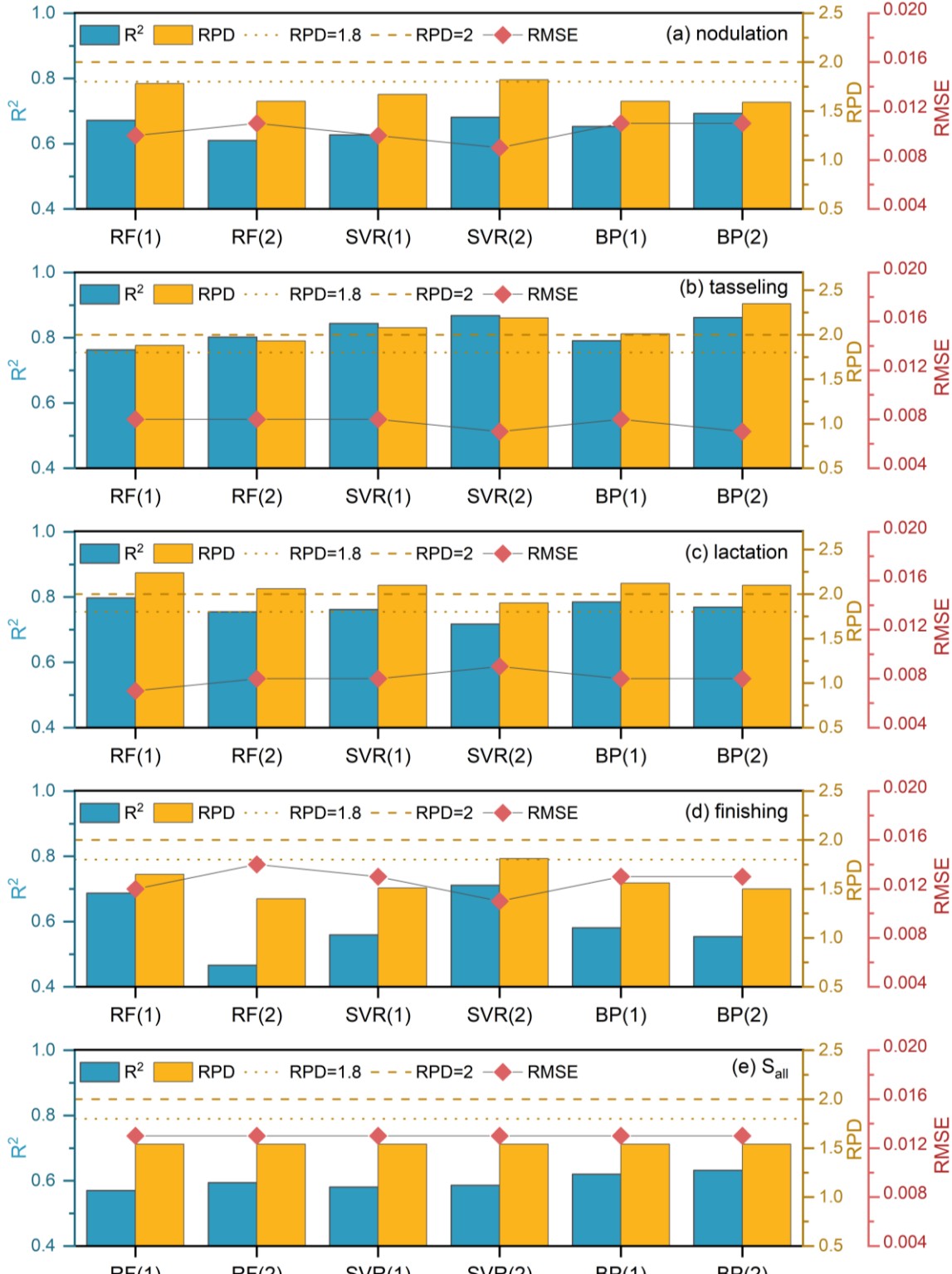

**Figure 11.** Model accuracy parameters for each growth stage. (**a**) represents the nodulation stage; (**b**) represents the tasseling stage; (**c**) represents the lactation stage; (**d**) represents the finishing stage; (**e**) represents the $S_{all}$ stage; (1) represents the model parameters $FD(R_\lambda)$ + OS_VI$_C$ + FD_VI$_C$ model using Random Forest: RF(1), Support Vector Regression: SVR(1) and Back Propagation Neural Network: BPNN(1); (2) represents the model parameters $FD(R_\lambda)$ + OS_VI$_C$ + VI$_S$ model using Random Forest: RF(2), Support Vector Regression: SVR(2) and Back Propagation Neural Network: BPNN(2). BP in the figure represents BPNN.

## 4. Discussion

### 4.1. Effect of FD on Anth Estimation

FD were obtained by the spectral conversion of OS of maize leaves in the 380–1000 nm band range. OS and FD were correlated with Anth and were used to construct Anth-UR. At all growth stages of maize, the maximum values of the correlation coefficients between FD and Anth and the FD($R_\lambda$)_based Anth-UR were improved compared to the OS. For the nodulation, tasseling, lactation, and finishing stages, the correlation coefficients of the FD increased by 11.48%, 14.09%, 9.46%, 45.12%, and 27.94%, respectively. Additionally, the $R^2$ increased by 28.6%, 44.8%, 11.3%, 82.1%, and 73.2%, while the RMSE decreased by 16.5%, 21.2%, 6.8%, 12.3%, and 15.8%, and the RPD improved by 19.7%, 26.9%, 7.3%, 14.0%, and 18.7%. Previous studies have shown that using FD can improve the correlation with the study target and enhance the modeling accuracy. For instance, Li et al. used hyperspectral canopy techniques to study N concentrations in winter oilseed rape leaves [53], Zhang et al. used hyperspectral inversion of soil heavy metals [54], and Liu et al. used UAV hyperspectral techniques to estimate potato biomass [14]. These studies demonstrated that FD could effectively highlight the location of the spectrally sensitive bands and enhance the degree of spectral response to the studied target. This demonstrates that first-order spectral conversion can effectively highlight the location of spectrally sensitive bands and enhance the spectral response to the studied target. This is because the FD can remove the influence of background noise on the target and refine the spectral information.

### 4.2. Effect of VI$_S$ on Anth Estimation

The performance of R/G for Anth estimation varied greatly at different growth stages of maize. The accuracy of the Anth-UR model ($R^2 > 0.6$) was better for R/G at the tasseling stage but poor for the rest of the growth stages. ACI, MACI, ARI, and MARI performed better than R/G in other growth stages, which is in agreement with Gitelson's study on anthocyanin estimation in hazel and maple leaves [28]. The principle of R/G index construction is to estimate Anth by the ratio of the red spectral region, where only chlorophyll is absorbed, to the green spectral region, where both chlorophyll and anthocyanin are absorbed [24]. In contrast, ACI, MACI, ARI, and MARI account for the effect of blade structure by incorporating the near-infrared band or the red-edge band into their calculations.

In addition, the accuracy of the Anth-UR models constructed by ACI and MACI was overall higher than by ARI and MARI in this study. This is not consistent with the results of Steele and Gitelson [26]. The main reason may be the inconsistency in the selected range of the calculated wavebands. Specifically, the red-edge band range was selected as 700–760 nm based on the spectral curve characteristics in this paper, while the range selected by Gitelson and Steele was 690–710 nm. Additionally, the correlation coefficients between Anth and VI$_S$ constructed based on the red-edge band (700–760 nm) were found to be higher than those constructed based on the red-edge band (690–710 nm) in this study. Nevertheless, the accuracy of the Anth-UR model constructed based on ACI or MACI is still not sufficient for the quantitative estimation of anthocyanins (RPD < 1.8). Therefore, the VI$_S$-based Anth-MR model is constructed in this paper. The results showed that the accuracy of the VI$_S$-based SVR model was satisfactory, with RPD values of 1.81 at the nodulation stage and 1.95 at the tasseling stage, respectively. Additionally, the VI$_S$-based BPNN model exhibited a moderate RPD of 1.86 at the tasseling stage. The superiority of SVR models for estimating anthocyanins was demonstrated earlier in a study by Qin et al. for nondestructive estimation of anthocyanins in grape leaves [55]. Therefore, using the VI$_S$-based SVR model to estimate anthocyanins at the nodulation and tasseling stages of maize is a good choice.

Based on the characteristics of multiple and narrow hyperspectral bands, any two-band combination of OS and FD in the range of 380–1000 nm was performed to find the optimal band combination for constructing NDVI, RVI, and DVI. In the present study, the correlation between VI and Anth was analyzed, and the descending order of VI and Anth correlation relationships at each growth stage was: FD_VI$_C$ > OS_VI$_C$ > VI$_S$. The results

indicate that the VI constructed from the FD had the strongest correlation with Anth at all growth stages, followed by the VI constructed from the OS and the $VI_S$, respectively. Overall, the order of accuracy of Anth-UR models constructed based on these three types of VI was: $VI_C > VI_S$. The Anth-UR models constructed based on the OS or FD had different performances at different growth stages. Moreover, the Anth-UR models based on OS_DVI, FD_DVI, FD_RVI, and FD_NDVI demonstrated satisfactory accuracy for anthocyanin estimation at the tasseling stage, with RPD values exceeding 1.8. Similarly, at the lactation stage, the Anth-UR models based on FD_DVI and FD_NDVI also exhibited RPD values exceeding 1.8, indicating their potential for quantitative estimation of anthocyanins. This reflects the superiority of NDVI, RVI, and DVI based on the optimal band combination constructed by any two bands. Tanaka et al. conducted a study where they compared the optimal NDVI, RVI, and DVI, constructed based on any two bands, with existing vegetation indices such as NDVI, EVI, and OSAVI. Their findings demonstrated that the UR model based on the optimal DVI exhibited superior accuracy in estimating the leaf area index of winter wheat [30]. These results align with the findings presented in this paper.

Additionally, Anth-MR models were constructed for each growth stage of maize using three different combinations: $FD(R_\lambda) + OS\_VI_C$, $FD(R_\lambda) + FD\_VI_C$, and $FD(R_\lambda) + VI_S$. Surprisingly, the $FD(R_\lambda) + VI_S$-based Anth-MR model performed well, with potentially higher accuracy at the nodulation and tasseling stages compared to the other two models. This may be because the $VI_S$ index uses more spectral bands than the $VI_C$ index, and the $VI_C$-based Anth-MR models may have neglected some information associated with Anth. This needs to be further investigated in future research. Furthermore, Anth-MR models were constructed using $FD(R_\lambda) + OS\_VI_C + VI_S$ and $FD(R_\lambda) + OS\_VI_C + FD\_VI_C$ as model inputs, respectively, and it was found that the model accuracy of $FD(R_\lambda) + OS\_VI_C + VI_S / FD\_VI_C$ was higher overall than that of $FD(R_\lambda) + OS\_VI_C / FD\_VI_C / VI_S$. In the study by Lili Luo et al., Anth-UR and Anth-MR models were constructed based on $R_\lambda$, $VI_S$, $VI_C$, and $R_\lambda + VI_S + VI_C$. It was found that the $R_\lambda + VI_S + VI_C$-based Anth-MR model performed better than models constructed based on the other three spectral parameters [31]. This is consistent with the results of this paper.

### 4.3. Effect of Different Growth Stages on Anth Estimation

In this study, anthocyanin estimation models were constructed at critical growth stages of maize: nodulation, tasseling, lactation, and finishing. The highest precision of anthocyanin estimation models differed at different growth stages. For nodulation and tasseling, the highest accuracy was achieved by the $FD(R_\lambda) + OS\_VI_C + VI_S$-based SVR model, with $R^2$ of 0.68 and 0.87, RMSE of 0.007 and 0.007, and RPD of 1.82 and 2.19 for the validation set, respectively. For lactation, the highest accuracy was obtained for the $FD(R_\lambda) + OS\_VI_C + FD\_VI_C$-based RF model with $R^2$ of 0.80, RMSE of 0.007, and RPD of 2.24 for the validation set. For finishing, the highest accuracy was obtained for the $FD(R_\lambda) + OS\_VI_C + VI_S$-based SVR model with $R^2$ of 0.71, RMSE of 0.011, and RPD of 1.81 for the validation set. For the $S_{all}$ stage, the highest accuracy was obtained for the $FD(R_\lambda) + OS\_VI_C + VI_S$-based BPNN model with $R^2$ of 0.63, RMSE of 0.013, and RPD of 1.54. Obviously, the highest accuracy of the anthocyanin estimation model was higher at the tasseling and lactation stages than at the nodulation and finishing stages. Additionally, it was found that with the same modeling approach and parameter types, the model accuracy was overall higher for the tasseling and lactation stages than for the nodulation and finishing stages. This is not a coincidence and is an indication that different growth stages do have an effect on anthocyanin estimation in maize leaves. Qin et al. found different sensitive bands associated with Anth at different growth stages, which has implications for anthocyanin estimation in grape leaves [55]. These findings are consistent with the results of this study.

### 5. Conclusions

This study was to estimate the Anth of maize leaves at individual and whole growth stages based on hyperspectral data of maize leaves at 380–1000 nm, combined with ma-

chine learning (ML) methods. In this paper, we measured the hyperspectral data and the corresponding relative anthocyanin content data of maize leaves at the critical growth stages of nodulation, tasseling, lactation, and finishing of maize. FD were derived from the original spectra OS. First, the spectral parameters highly correlated with Anth were selected based on the correlation between different spectral reflectance and Anth. A total of two sensitive bands ($OS(R_\lambda)$ and $FD(R_\lambda)$), five classical vegetation indices (R/G, ACI, MACI, ARI, and MACI), and six optimized vegetation indices (OS_NDVI, OS_RVI, OS_DVI, FD_NDVI, FD_RVI, and FD_DVI) were selected from OS and FD. Then, 13 Anth-UR models were constructed based on each of these parameters at different growth stages of maize. The effects of these 13 parameters on the accuracy of the Anth-UR models were analyzed and compared. As the Anth-UR models were not stable, several Anth-MR models were also constructed. The combination of two types of spectral parameters ($FD(R_\lambda) + VI_S/OS\_VI_C/FD\_VI_C$) or the combination of three types of spectral parameters ($FD(R_\lambda) + OS\_VI_C + FD\_VI_C/VI_S$) were used as inputs to the models. The main conclusions of this study are as follows:

(1) FD can effectively highlight the location of spectrally sensitive bands and enhance the degree of spectral response to the study target. The first-order spectral conversion could effectively improve the correlation between $R_\lambda$, $VI_C$, and Anth, and $VI_C$ are usually more sensitive to Anth than $VI_S$;

(2) The performance of Anth estimation models constructed based on different vegetation indices varied. The accuracy of the Anth-UR models based on $VI_C$ was higher than that based on $VI_S$. The overall performance of Anth-MR models was better than that of Anth-UR models. The highest accuracy of the Anth-MR models was obtained with the $FD(R_\lambda) + OS\_VI_C + VI_S/FD\_VI_C$ models;

(3) There are effects of different growth stages on Anth estimation models. The tasseling stage and lactation stage were found to be better growth stages for estimating Anth in maize leaves. For the tasseling stage, the best model was the $FD(R_\lambda) + OS\_VI_C + VI_S$-based SVM model, with an $R^2$ of 0.868, RMSE of 0.007, and RPD of 2.19. For the lactation stage, the best-performing model was the $FD(R_\lambda) + OS\_VI_C + FD\_VI_C$-based RF model, with an $R^2$ of 0.797, RMSE of 0.007, and RPD of 2.24.

**Author Contributions:** Conceptualization, S.J.; methodology, S.J.; software, S.J.; validation, S.J., X.W., Z.Z. and Q.C.; formal analysis, S.J.; investigation, S.J., Z.Z., Q.W. and Y.Z.; resources, Q.C.; data curation, S.J., Z.Z., Q.W. and Y.Z.; writing—original draft preparation, S.J.; writing—review and editing, S.J., X.W. and Q.C.; visualization, S.J.; supervision, X.W. and Q.C.; project administration, Q.C.; funding acquisition, Q.C. All authors have read and agreed to the published version of the manuscript.

**Funding:** This research was funded by the National High Technology Research and Development Program of China (863 Program), grant number 2013AA102401-2.

**Data Availability Statement:** Data sharing is not applicable to this article.

**Acknowledgments:** We would like to thank all the students in Chang's team for collecting the data for us.

**Conflicts of Interest:** The authors declare no conflict of interest.

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
