# Peer review of "Estimation of Anthocyanins in Whole-Fertility Maize Leaves Based on Ground-Based Hyperspectral Measurements"

_remotesensing, doi:10.3390/rs15102571_

Round 1

Reviewer 1 Report

1. The left column of the first page: “Citation:Jiang, S.”. Modification: add a space after the colon, “Citation: Jiang, S.”.

2. Line 68: “and VIC”. Modification: add a comma “,” before “and”, “ and VIC”.

3. Line 75: “in low accuracy of the models”. Modification: “in the low accuracy of the models”.

4. Line 85: “nondestructive and”. Modification: add a space after the colon, “nondestructive, and”

5. Line 128: “Determination of Relative Anthocyanin content”. Modification: “content” first letter needs to be capitalized, “Determination of Relative Anthocyanin Content”.

6. Line 131: “real-time”. Modification: “real time”.

7. Line 254: “and and VIS”. Modification: Delete a redundant “and”, “and VIS”.

8. Line 320: “Figures 7”. Modification: “Figure 7”.

9. Line 447: “SVR model”. Modification: “SVR models”.

10. Line 513: “the overall”. Modification: “The overall”.

Minor editing of English language required

Author Response

Thank you for your valuable advice, which is very useful for my manuscript. I have tried my best to respond to your suggestions point by point, please see the attachment for details. I hope my reply will satisfy you and thank you again.

Reviewer 2 Report

The authors used the hyperspectral reflectance of maize leaves and the corresponding Anth indicators for modeling and discussed the relationship between Anth indicators and spectral information during the key growth stages of maize, such as setting, tasselling, lactation and harvesting. While I am not an expert in the fields of agricultural remote sensing and hyperspectral remote sensing, etc., I offer the following suggestions from other perspectives, such as machine learning, for the authors' reference.

1. From the introduction section, the data acquisition (referring to hyperspectral data acquisition) and modeling section of this study lack the corresponding theoretical support. It is suggested that the authors cite related theories and literature to provide theoretical knowledge about the anthocyanins estimation methods based on hyperspectral information.

2. the authors utilized three methods for modeling, which are RF, SVR, and BPNN, but the authors did not specify the training parameters for the three models. This may affect the understanding of other readers and the replication of the method in other regions for that method. I suggest that the authors further supplement the parameter descriptions in the method section. It would be better to open source all the data and code.

3. In general, there is a lack of corresponding literature support, especially the research on hyperspectral remote sensing images related to agricultural remote sensing. It is suggested that the authors enhance the relevant research section.

4. The authors collected data from 80 sample points as research data, whether there is a relevant public dataset or more data for training and validation, the research sample is small and not convincing from a machine learning perspective. Moreover, XGBoost has a good performance for machine learning tasks with small samples. Why not try more models for this task? 

5. I suggest that the authors add the following information to the Conclusion section, including a brief description of the main objective of the research, as well as the methods and the data used. 

Minor editing of description is required.

Author Response

(The authors gave the same response as above.)

Reviewer 3 Report

The paper "Estimation of Anthocyanins in Whole-Fertility Maize Leaves Based on Ground-based Hyperspectral Measurements" estimated anthocyanins in maize leaves at different growth stages based on remote sensing methods. It is shown that using first-order differential spectra and models using optimized vegetation indices benefits the accuracy of estimating anthocyanins in maize leaves. The paper also justifies which maize growth stages the best results are obtained using the proposed models.

Interesting conclusions are drawn, with possible applications in the agricultural modelling, although they suggest some changes and some questions arise for this reviewer:

1) In line 97, it is indicated that six experiments have been carried out for each treatment, whereas in Table 1, for N, only five experiments are shown.

2) In section 2.2 and linking it with the previous section, it would be convenient to add images of the data collection, at least one of each growth phase, so that the reader can check the phenological state of the plant.

3) In section 2.2.1, adding an image of the device used would be helpful to make it easier for the reader to identify it. If this is not possible, a reference explaining and showing the device or a web page should be added.

4) Indicate on lines 124-125 which software was used to resample to 1nm the spectrum obtained with the SVC HR-1024i device. An image showing the spectrum before and after smoothing should be added.

5) The text states that MATLAB R2019b has been used (lines 169, 175, 183, ...). A reference should be added to the bibliography.

6) In section 3.3.1, it is stated on line 246 that the correlation between Anth and FD in absolute value for tasselling is 0.825 at 524 nm, but in the graph, in Figure 3(b), it looks like it could also be close to 600 nm at 580 nm. Explain the choice that has been made.

7) Figure 5 has to be explained in the caption how the figure is constructed as the phases depicted on the right-hand side are overlooked (each growth stage). The same for Figure 6

8) The reason for obtaining Contour maps of the Correlation coefficient between FD_VIC and Anth with white colours has to be added in the paragraph from lines 282 to 288. Please explain in the text what has happened so that it does not correspond to the colour bar in the inset of the figure.

9)  Figures 8 and 9 have to change BP to BPNN or indicate in the caption that a new nomenclature for BPNN is being used.

10) In the text or the figure caption (Figures 8 and 9), it would be good to remind the reader of the meaning of the acronyms RF, SVR, and BPNN. For example             

(1) represents the parameters of the FD(Rλ) + OS_VIC model using Random Forest: RF(1), Support Vector Regression: SVR(1) and Back Propagation Neural Network: BPNN(1).

11) Reference [4] indicates Chalker-Scott* but should indicate Chalker-Scott

12) Substitute in reference [7] Close, D. C.; Beadle, C. L. for CLOSE, D.C.; BEADLE, C.L.

Author Response

(The authors gave the same response as above.)

Round 2

Reviewer 2 Report

The paper has been greatly improved. I appreciate the revision of additional descriptions compared to the previous version. In its current state, the paper is much easier to follow. Conditional on a few minor comments I recommend accepting the paper and congratulating the authors on developing a useful interdisciplinary approach by integrating remote sensing and agriculture.

1. Please standardize the use of parentheses, e.g. on page 1, line 13 'The estimation of anthocyanin(Anth)' (note: no space before the parentheses). However, on page 1, line 18 'the original spectra (OS).' , (space before the brackets). Please ask the authors to check the whole text and unify whether spaces need to be inserted.

2. The styles of Table 1 and Table 2 (as well as Table 3) are different, so please confirm and unify them.

3. Both two graphs in Figure 4 are missing the title at the top. The same as Figures 5, 6, 9, 10, and 11. 

4. The bracket symbols in line 187 do not look like the same font, please check it again. The same as the symbol ‘℃' in line 130. 

5. For me, the description of the parameters in sections 2.3.3, 2.3.4 and 2.3.5 is not particularly clear. It would be beneficial for the authors to elaborate on the setting of the parameters to reproduce the method.

Author Response

Thank you very much for giving us a chance to revise and for your suggestions. Please see the attachment for our response to your suggestions in detail.
